# Evaluation of a Florfenicol Formulation for Treating Streptococcosis and Francisellosis in Nile Tilapia (*Oreochromis niloticus*): A Study of Safety, Withdrawal Period and Efficacy

**DOI:** 10.3390/microorganisms13030558

**Published:** 2025-03-01

**Authors:** Natália Amoroso Ferrari, Raffaella Menegheti Mainardi, Mayza Brandão da Silva, Gabriel Diogo Guimarães, Marcos Letaif Gaeta, Francisco Eduardo Pereira Rocha, Tainara Oliveira da Silva, Alene Santos Souza, Bruna Cordeiro Baptista, João Anderson Keiti Rocha, Erika Fernandes Lopes Maturana, Daniela Dib Gonçalves, Nelson Maurício Lopera Barrero, Giovana Wingeter Di Santis, Ulisses de Pádua Pereira

**Affiliations:** 1Laboratory of Fish Bacteriology, Department of Preventive Veterinary Medicine, State University of Londrina, Londrina 86057-970, Paraná, Brazil; natalia.amoroso@uel.br (N.A.F.); raffaella.menegheti@uel.br (R.M.M.); mayza.brandao.silva@uel.br (M.B.d.S.); gabriel.diogo.guimaraes@uel.br (G.D.G.); mlgaeta@uel.br (M.L.G.); edu13rocha@gmail.com (F.E.P.R.); ttainaraoliveira.s@hotmail.com (T.O.d.S.); alenesantos47@gmail.com (A.S.S.); 2SAN Group Biotech Brazil Ltda., Campinas 13058-009, São Paulo, Brazil; bruna.baptista@san-group.com (B.C.B.); joao.rocha@san-group.com (J.A.K.R.); 3Laboratory of Animal Pathology, Department of Preventive Veterinary Medicine, State University of Londrina, Londrina 86057-970, Paraná, Brazil; lopesmaturana@gmail.com (E.F.L.M.); giovanaws@uel.br (G.W.D.S.); 4Department of Preventive Veterinary Medicine and Public Health, Paranaense University, Umuarama 87502-210, Paraná, Brazil; danieladib@unipar.br (D.D.G.); 5Department of Animal Science, State University of Londrina, Londrina 86057-970, Paraná, Brazil; nmlopera@uel.br (N.M.L.B.)

**Keywords:** aquaculture, antibiotic, *Streptococcus agalactiae*, *Francisella orientalis*

## Abstract

This study evaluates the efficacy of a new florfenicol-based drug, both in vitro and in vivo, in Nile tilapia (*Oreochromis niloticus*) against pathogens commonly found in fish farming and its withdrawal period. The drug’s efficacy was tested using prophylactic, metaphylactic, and therapeutic approaches against *Streptococcus agalactiae* (serotypes Ib and III) and *Francisella orientalis*. The minimum inhibitory concentration of florfenicol was 4, 5, and 128 μg/mL for the different pathogens. Fish mortality was lower in the prophylactic treatment compared to the metaphylactic and therapeutic treatments for *S. agalactiae*. No difference in mortality was observed for *F. orientalis* across treatments. Mild to moderate lesions suggestive of intoxication were observed, mainly in the liver of fish treated with overdoses or exposed to low temperatures. Florfenicol reduced mortality rates, especially with early treatment (metaphylactic), in fish experimentally challenged with the pathogens. Moreover, prophylactic antimicrobial use is not recommended, as it promotes the selection of multidrug-resistant bacterial strains. Additionally, the residual concentration of the drug in muscle tissue lasted for a shorter period than that recommended by the manufacturer, and at lower concentrations than required by national and international legislation.

## 1. Introduction

One of the greatest challenges in fish aquaculture is the occurrence of bacterial disease outbreaks, which can result in production losses of up to 80% and pose significant health risks to the industry [1]. The major bacterial agents responsible for these outbreaks include *Streptococcus* spp., *Francisella orientalis*, *Edwardsiella* spp., and *Aeromonas* spp. [2]. One of the main strategy for controlling outbreaks with high mortality has been the use of antimicrobials, particularly for bacterial diseases that are not yet preventable through immunoprophylaxis [3].

Although there are several commercial formulations of antimicrobials for veterinary use, only florfenicol and tetracycline are approved for use in aquaculture in Brazil [4]. In this context, studies indicate that the prolonged use of these molecules can favor the increase in resistant bacteria, a phenomenon which motivates the search for more effective and safer formulations with a rapid withdrawal to minimize such impacts and comply with international food safety standards [5].

Florfenicol is a synthetic antimicrobial agent belonging to the amphenicol group with a broad-spectrum bacteriostatic action [6]. In veterinary medicine, it treats infections caused by Gram-positive and Gram-negative bacteria resistant to chloramphenicol and thiamphenicol [7]. The molecule’s mechanism of action involves inhibiting peptide bond formation in the prokaryotic 70S ribosome, thereby disrupting protein synthesis. Its superior efficacy compared to other antimicrobials in the same class is due to the substitution of a p-nitro group with a sulfomethyl group, which prevents inactivation by acetyltransferase enzymes responsible for the resistance to chloramphenicol and thiamphenicol [8], making it more effective in treating bacterial infections in animals [7]. Additionally, florfenicol is considered safer than chloramphenicol, as it is not associated with severe adverse effects, such as idiosyncratic aplastic anemia, which can occur in humans [6]. Furthermore, the antibiotic used to treat fish must be effective in controlling the disease and must not remain at a high concentration at the time of slaughter, particularly in the muscles and skin, which are the most consumed parts by humans [9].

To ensure food safety, it is recommended that the use of florfenicol be discontinued 15 days before the slaughter of freshwater fish. However, some manufacturers indicate that four days may be sufficient for the drug concentration to fall to acceptable levels [10]. Therefore, new formulations should be evaluated to verify their effectiveness and safety according to One Health risk settings [11].

Florfenicol is an important antimicrobial agent for controlling bacterial outbreaks in fish, including those caused by *Francisella* species. Studies have shown that tilapia treated with florfenicol exhibited survival rates ranging from 86.7% to 100% when treatment was initiated one to three days after the experimental challenge [12]. Additionally, the drug reduced mortality in fish challenged with *Streptococcus iniae*, decreasing from 35.8 ± 4.4% in untreated individuals to 2.5 ± 1.1% in those treated with 15 mg/kg four hours after inoculation [13]. These results were obtained from tests with different drug brands. Despite the evidence of its efficacy, many of these studies are outdated, highlighting the need for new validations. This necessity arises from the fact that bacterial populations undergo constant changes due to the selective pressure driven by antimicrobial use. Although several studies have already evaluated the efficacy of florfenicol in fish, most were conducted on bacterial populations from previous periods and did not take temperature variations into account. This factor is crucial, as the drug’s metabolism in the fish’s body can be influenced by temperature, which may impact its withdrawal throughout the seasons. Beyond efficacy, florfenicol has demonstrated a favorable safety profile in biosafety studies conducted on different fish species. When administered orally, it exhibits rapid absorption, wide distribution, and a good physiological response [11]. Finally, some brands of florfenicol that previously produced the drug are no longer available on the market, creating a gap and generating demand for new florfenicol-based products intended for aquaculture which must be properly validated.

This study aims to evaluate the efficacy of a florfenicol formulation against *Streptococcus agalactiae* serotypes Ib and III and *Francisella orientalis* in Nile tilapia (*Oreochromis niloticus*), administered orally mixed with feed through prophylactic, metaphylactic, and therapeutic approaches. Additionally, the quantities of drug residues in the muscle of fish treated with florfenicol are analyzed, observing the depletion curve of residues and/or metabolites.

## 2. Materials and Method

### 2.1. Ethics Committee on Animal Use

All animal procedures were approved by the Ethics Committee on Animal Use of the State University of Londrina (Approval number CEUA-UEL OF.020.2022). All experiments, including the sample size, follow the International Cooperation on Harmonization of Technical Requirements for Registration of Veterinary Medicinal Products (VICH) relevant guidelines and regulations [14].

### 2.2. Bacterial Strain

In this study, three previously isolated and identified bacterial strains were used in the experiments: *Streptococcus agalactiae* serotype Ib, strain S13 [15] and serotype III, strain S73 [16], both cultured on Mueller Hinton agar (Himedia, Maharashtra, India) enriched with 5% of defibrinated sheep blood and incubated at 28 °C for 48 h, as well as *Francisella orientalis*, strain F1 [17], which was grown on cystine–heart agar supplemented with 1% bovine hemoglobin and incubated at 28 °C for 96 h. All bacterial strains were originally isolated during outbreaks in Nile tilapia.

### 2.3. Minimum Inhibitory Concentration

The active ingredient, florfenicol, was evaluated using the microdilution method against *S. agalactiae* serotypes Ib and III and against *F. orientalis*. The tests were conducted at two concentration ranges, i.e., 1× (128 µg/mL to 0.062 µg/mL) and 2.5× (320 µg/mL to 0.156 µg/mL), to determine the minimum inhibitory concentration (MIC) and minimum bactericidal concentration (MBC), following CLSI guidelines [18]. Measurements were recorded after 24 h of incubation.

### 2.4. Fish

A total of 4690 clinically healthy juvenile male Nile tilapia were sourced from a commercial hatchery in Paraná State, Brazil. Of these, 1300 fish with a mean weight of 60.5 g were used for the efficacy test, while 3390 fish with a mean weight of 32.45 g were allocated for the residual limit test. The fish were housed in water tanks under optimal conditions for a 15 day acclimation period before the experimental challenge. These conditions were maintained throughout the experiment, with the water parameters as follows: a pH of 6.8–7.2, total ammonia <1 ppm, a chlorine content of 0 ppm, and a temperature of approximately 21 °C for the experiments involving *F. orientalis* and 28 °C for those with *S. agalactiae*. All chemical parameters of the water were measured using commercial kits (Alcon, Camboriu, Santa Catarina, Brazil), while the temperature was measured with thermometers.

The inclusion criteria for the animals in the study were as follows: clinically healthy fish in good health and nutritional condition and with zootechnical, clinical, and microbiological parameters within normal limits (assessed through euthanasia of a sample for evaluation). Additionally, the animals showed no behavioral changes at the time of selection. No animals were excluded from the study.

### 2.5. Medicated Feed

To prepare the medicated food, a feeding rate of 3% of the live weight of feed per day was calculated. The product, with a purity of 98.8%, total impurities of 0.56%, a water content of 0.07%, and a 100% active fraction (CVP 2015) (AMPHENOR^®^ 50; SAN Group Biotech Brasil Ltda., Campinas, Brazil), was incorporated into the feed at different concentrations (10 mg/Kg and 30 mg/Kg), with 5% (*w*/*v*) of the universal binding agent (carboxymethylcellulose-based; Vansil, Descalvado, SP, Brazil). Since the active ingredient concentration in the product is 50%, 0.67 g/kg of feed was added for the recommended dose and 2.01 g/kg of feed for the overdose. The product and the universal vehicle were homogenized using a mixer to perform the mixing process. Afterward, the mixture was sprayed onto the feed using a sprayer, then manually mixed and left to dry at room temperature.

### 2.6. Experimental Design

Two in vivo experiments, an efficacy test, and a residual and safety test were designed to evaluate the florfenicol-based product. The animals were randomly distributed in the tanks for all experiments; however, the distribution of the treatments could not be random due to the proximity between the tanks, in the case of the efficacy experiment. For this reason, the tanks that received the same experimental challenge were placed close together, while the different challenges were located far from each other.

#### 2.6.1. Efficacy Test

For the efficacy test, 50 fish per group, each with a mean weight of 60.5 g, were housed in 150 L tanks. Three treatment strategies were evaluated: prophylactic (the antibiotic administration began before the onset of clinical signs), metaphylactic (the antibiotic administration began when the first animal exhibited clinical signs), and therapeutic (the antibiotic administration began after at least three animals had shown clinical signs). These treatments were tested against three pathogens: *S. agalactiae serotypes* Ib and III and *F. orientalis*. All fish received the recommended florfenicol dose of 10 mg/kg, as suggested by the manufacturer. Mortality was monitored daily.

For the experimental challenge, *S. agalactiae* strains were cultured in BHI broth (Himedia, Maharashtra, India), and *F. orientalis* was cultured in Eugon broth (BD, Franklin Lakes, NJ, USA). The cultures were diluted in a saline solution to reach the following concentrations: 4 × 10⁶ CFU/mL for *S. agalactiae* serotype Ib, 2 × 10⁷ CFU/mL for *S. agalactiae* serotype III, and 1 × 10⁷ CFU/mL for *F. orientalis*. These concentrations were determined by bacterial counting. Each fish received a 0.1 mL intraperitoneal injection of the respective bacterial inoculum. The negative control group was injected with 0.1 mL of saline solution.

The experimental groups were organized as follows, with two groups for each treatment:-NC—non-infected fish feed with non-medicated food.-PC-SI—fish infected with *S. agalactiae* serotype Ib and fed with non-medicated food.-PC-S3—fish infected with *S. agalactiae* serotype III and fed with non-medicated food.-PC-FO—fish infected with *F. orientalis* and fed with non-medicated food.-PT-SI—fish infected with *S. agalactiae* serotype Ib and prophylactically fed.-PT-S3—fish infected with *S. agalactiae* serotype III and prophylactically fed.-PT-FO—fish infected with *F. orientalis* and prophylactically fed.-MT-SI—fish infected with *S. agalactiae* serotype Ib and metaphylactically fed.-MT-S3—fish infected with *S. agalactiae* serotype III and metaphylactically fed.-MT-FO—fish infected with *F. orientalis* and metaphylactically fed.-TT-SI—fish infected with *S. agalactiae* serotype Ib and therapeutically fed.-TT-S3—fish infected with *S. agalactiae* serotype III and therapeutically fed.-TT-FO—fish infected with *F. orientalis* and therapeutically fed.

At the end of the bacterial challenge, the remaining fish were euthanized using a benzocaine overdose (200 mg/L). Brain, liver, and spleen samples were collected and plated on agar (as described in Section 2.2) to identify the bacterial pathogen carriers. Additionally, all fish were examined for macroscopic lesions.

#### 2.6.2. Residual and Safety Test

A residual and muscular concentration test was conducted to determine the appropriate withdrawal period. For this purpose, 678 fish per group, with a mean weight of 32.45 g, were housed in 1000 L tanks and administered the recommended florfenicol dose (10 mg/kg). The test was performed at two different water temperatures, 21 °C and 28 °C, to evaluate potential differences in the molecule’s metabolism influenced by temperature. The fish were divided into two groups, each maintained at one of the specified temperatures.

Additionally, an overdose treatment (30 mg/kg) was administered to assess the occurrence of potential clinical signs of intoxication in fish at both water temperatures. Observations included an increased opercular movement frequency, gasping behavior, melanosis, and morphological changes [19].

Therefore, the experimental groups were as follows:-NC—fish fed with non-medicated food.-RL—fish fed with medicated food in the recommended dose in a low-temperature environment.-RH—fish fed with medicated food in the recommended dose in a high-temperature environment.-OL—fish fed with medicated food with an overdose in a low-temperature environment.-OH—fish fed with medicated food with an overdose in a high-temperature environment.

To perform the residual analysis, 10 fish from the RL and RH groups that received the recommended dose were euthanized in an ice bath and sampled on days 1, 5, and 10 of the treatment, as well as on days 1, 3, 5, 7, 10, and 15 post treatment. These samples were sent to an outsourced laboratory for the analysis of muscular residues of florfenicol and florfenicol amine.

Additionally, fish sampled on days 5 and 10 of the treatment, as well as on day 5 post treatment, had their liver, spleen, and gastrointestinal tract fixed in 10% buffered formalin for histopathological analysis to evaluate potential intoxication-related lesions. Fixed tissue samples were embedded in paraffin at 60 °C to produce cross-sections of 5 µm thickness which were stained with hematoxylin–eosin. The slides (Entellan, Darmstadt, Germany) were prepared and analyzed microscopically. A score was assigned to each lesion class following an adaptation from a previous study [20]. Detailed criteria and scoring descriptions are provided in the Appendix A.

### 2.7. Statistical Analysis

All statistical analyses in this study were conducted using PAST v.4.5 [21]. For the histopathological data, the sum of the lesion scores for each organ was initially analyzed. In cases where statistical differences were identified, the scores for individual lesions were compared separately.

Outliers were identified and removed using boxplots, and data normality was assessed with the Shapiro–Wilk test. For data with a normal distribution (*p* > 0.05), analyses were performed using an ANOVA followed by a Tukey’s post-hoc test for pairwise comparisons. For non-parametric data, the Kruskal–Wallis test was applied, followed by a Dunn’s test for pairwise comparisons. A 95% confidence interval was adopted.

## 3. Results

### 3.1. Minimum Inhibitory Concentration

The MIC of the florfenicol found was different for each bacterium tested, with 4 μg/mL for *S. agalactiae* serotype Ib, 5 μg/mL for serotype III, and 128 μg/mL for *F. orientalis*. Also, the MBC of the molecule was estimated at 8 μg/mL for *S. agalactiae* serotype Ib, 128 μg/mL for serotype III, and 128 μg/mL for *F. orientalis*.

### 3.2. Efficacy Test

In the susceptibility test, clinical signs such as exophthalmos, corneal opacity, and erratic swimming were observed starting on day 1 post infection in the PC-SI, MT-SI, and TT-SI groups, on day 2 in the PT-SI, PC-S3, and MT-S3 groups, on day 3 in the TT-S3 group, and only on day 14 in the PT-S3 group. For the group inoculated with *F. orientalis*, the only observed change was hyporexia on day 1.

Fish were monitored for 30 days, and florfenicol treatments were initiated at different time points post infection: the prophylactic treatment began on day 0 for groups inoculated with all three pathogens, the metaphylactic treatment started on day 1 for *S. agalactiae* serotype Ib and *F. orientalis* and on day 2 for *S. agalactiae* serotype III, and the therapeutic treatment was initiated on day 2 for *S. agalactiae* serotype Ib and *F. orientalis* and on day 3 for *S. agalactiae* serotype III.

Treated fish performed better than the positive controls for all pathogens tested. No significant differences were found among groups receiving the same treatment. Mortality was lower in the prophylactic treatment (21/100; 21%) compared to the therapeutic treatment (41/100; 41%) for *S. agalactiae* serotype Ib. For *S. agalactiae* serotype III, the prophylactic treatment showed the lowest mortality (2/100; 2%). No significant differences in mortality were observed among the treatments for *F. orientalis* (Table 1).

In the positive control groups, mortality rates were as follows: 88/100 (88%) for *S. agalactiae* serotype Ib, 61/100 (61%) for *S. agalactiae* serotype III, and 57/100 (57%) for *F. orientalis*. No mortality was observed in any of the negative control groups. A Fisher’s exact test revealed significant differences (*p* < 0.05) in mortality rates among treatments for *S. agalactiae* serotypes Ib and III.

The surviving fish were evaluated for macroscopic lesions during autopsy. Fish infected with *S. agalactiae* exhibited exophthalmos and meningeal congestion, and fish infected with *F. orientalis* presented spleen granulomas (Table 2). The fish organs were cultured during the autopsy to re-isolate the bacteria and determine the carrier state (Table 3).

### 3.3. Residual Test

No fish died during the 25-day observation period of the residual test. Mild hyporexia was observed in the groups kept at the low temperature (RL and OL) starting on day 1 of the treatment, becoming more pronounced by day 5, where food consumption was approximately 50% of the expected intake. No other signs of intoxication were observed in any of the groups, including both the recommended dosage and the overdose groups.

Animals sampled for residual muscular analysis had slightly higher values in the high-temperature water groups, with approximately 15,000 μg/kg (florfenicol + florfenicol amine—F + FA) after 10 days of treatment compared to 1250 μg/kg in the fish kept in the low-temperature water. A similar pattern was observed in fish treated with florfenicol alone, with a maximum value of 796 μg/kg in the low-temperature group versus 928 μg/kg in the high-temperature group (Table 4).

Histopathological analysis showed only mild inflammation in the stomach of all groups (*p* = 0.37) (Figure 1). In the spleen, there was a significant difference in congestion (*p* < 0.01), ranging from mild to moderate in the OH, OL, and RL groups on day 5 of the treatment. By day 10 of the treatment and day 5 post treatment, the scores for this lesion had decreased, ranging from absent to slight in all groups (Figure 2).

In the liver, a mild reduction in glycogen accumulation was observed (*p* < 0.01), mainly in the RL group and the negative control group on day 10 of the treatment. The same collection also showed mild necrosis (*p* < 0.01) in the RL and OL groups, with moderate necrosis observed in the OH group. On day 5 post treatment, RL fish showed mild necrosis, accompanied by inflammation ranging from mild to moderate (*p* < 0.01).

All other groups exhibited some level of inflammation, which was more pronounced in the RL group on day 5 of the treatment and in the RL and RH groups on day 5 post treatment (mild to moderate). A moderate presence of pigmented macrophages was noted in the RL group on day 5 of the treatment and in the OH group on day 10 of the treatment (*p* < 0.01). Mild congestion was observed in the OL and RH groups on day 5 of the treatment, in the OL group on day 10 of the treatment, and in the RH group on day 5 post treatment. Congestion varied from mild to moderate in the RL and OH groups on day 10 of the treatment and in the OL group on day 5 post treatment (*p* < 0.01) (Figure 3).

## 4. Discussion

Bacterial diseases are one of the biggest challenges to fish production [2]. Antibiotics remain the primary treatment method to control these diseases, particularly those caused by *Francisella orientalis*, for which no commercial vaccine is available [22].

In Brazil, only two antibiotics are approved for use in aquaculture: tetracycline and florfenicol [4]. A study analyzes 181 mortality outbreaks in fish farms across different Brazilian states and evaluates the resistance profile of 232 isolated bacteria. The resistance rate found for florfenicol was 9.48%, while, for tetracycline, it was 38.36% [23]. The authors highlight that this result is not surprising, as tetracycline is one of the most used antimicrobials in veterinary medicine and is often the drug of choice for producers during mortality outbreaks. While research into new molecules is crucial [24], it is also important to evaluate new formulations of approved molecules. This is essential to account for potential pharmacokinetic differences, bioavailability, purity of the active ingredient, excipients, and other components that may vary between products from different brands [25].

In this study, the administration of florfenicol-medicated feed resulted in a significant difference in mortality between infected fish and the control group. The survival rate for *F. orientalis* was 95% in the prophylactic treatment, 94% in the metaphylactic treatment, and 87% in the therapeutic treatment. This was higher than the 63% survival rate reported by Soto et al. [26], using the same 10 mg/kg fish dose. The efficacy of early treatments highlights the importance of trained professionals and continuous surveillance in aquaculture farms to quickly identify the need for antibiotic therapy. It should be noted that the prophylactic or inappropriate use of antimicrobials is not encouraged, but continuous monitoring by competent professionals is encouraged.

The survival rate for francisellosis was similar to that obtained by Favero et al. [20] with early treatment using oxytetracycline. However, differences in clinical signs, bacterial re-isolation, and MBC were observed. Furthermore, the effectiveness of an antibacterial treatment may offer a clinical cure but not completely eliminate the fish pathogens in tissues such as the spleen and kidney [26]. These differences in treatment efficacy are not entirely clear due to this pathogen’s facultative intracellular lifestyle, and more comprehensive studies, including field trials, must be performed.

The MIC of florfenicol against *S. agalactiae* serotype Ib was 4 μg/mL, while, for serotype III, it was 5 μg/mL. These values are lower than the MICs reported by Kalaria et al. [27] which ranged from 8 to 80 μg/mL. No significant difference (*p* = 0.73) was found between the mortality of the metaphylactic treatment (started on day 1) with florfenicol and the data by Faria et al. [28], who used early treatment (24 h after infection) with oxytetracycline. However, the recovery of the bacteria after the infection was significantly lower in the florfenicol treatment (37%) compared to oxytetracycline (75%). Costa et al. [23] found a significant difference in the antibiotic resistance of isolated *Streptococcus* spp. between florfenicol (1.2%) and tetracycline (11.7%). The combination of better bacterial clearance, lower antibiotic resistance, and similar survival rates in experimental infections suggests that the florfenicol formulation used in this study is more effective in treating streptococcosis than tetracycline.

Muscular residues of F + FA were below the maximum residue limit of 800 µg/kg, as required for export and compliance with international and national regulations [29], after just one day without treatment at both temperatures. However, similar to another study with tilapia, F + FA concentrations were higher at the higher temperature during treatment, suggesting that the absorption and distribution of the drug were faster in fish kept at the higher temperature compared to those at low temperature [30]. Even with a 30% safety margin, the estimated withdrawal period was less than two days after the end of the treatment. Kosoff et al. [31] found a 4 to 6 days withdrawal period in the muscle and skin of Nile tilapia, while the formulation used in this study had a withdrawal period of approximately half that time. These differences may be related to the composition of each formulation which, together with external factors, can interfere with the absorption, metabolization, and elimination time of the molecule [25].

Although F + FA concentrations were measured in tilapia muscle to verify the withdrawal period and safety for human consumption post treatment, concentrations in the liver and other tissues were not assessed. The liver, being responsible for metabolizing the drug and its compounds, showed mild to moderate lesions, particularly in the RL and OL groups, both of which were kept at the lower temperature of 21 °C. This suggests that the low temperature may have delayed the metabolization and depletion of florfenicol [31]. A study that evaluated the concentrations of florfenicol and its compounds in the muscle, liver, and kidneys of tilapia during and after the treatment at 28 ± 2 °C found higher concentrations in the kidneys, followed by the liver and fillet. This may explain why lesions were still observed in the liver even after F + FA levels had decreased post treatment [30].

Inflammation, reduction in glycogen accumulation, and necrosis were observed in Nile tilapia on the 10th day of the treatment with florfenicol at a dose of 15 mg/kg and a temperature range of 23 to 29 °C, which was also observed in our study [32]. The same study noted that the intensity of the lesions in the liver and kidneys was dose-dependent, a phenomenon which may account for the moderate congestion, necrosis, and presence of pigmented macrophages in the OH group. Despite the injuries found, none of the animals died during the observation period. Furthermore, it is known that the liver is an organ with regenerative capacity, which has already been demonstrated in other studies involving florfenicol in tilapia [33].

One of the main challenges in the approval of antimicrobial drugs for aquaculture is the high cost associated with generating the scientific data required for regulatory approval. These data typically include information on human food safety, target animal safety, efficacy, and environmental impact. Among these impacts, water contamination by antimicrobials can disrupt various ecosystems and affect multiple species, including humans [34]. The medication can leach from the feed into the water and be absorbed by aquatic organisms through exposure. [35]. Nevertheless, using these molecules is necessary and remains one of the primary measures for treating bacterial infections. For this reason, studies investigating these interactions and monitoring their responsible use are fundamental.

In this study, we provide valuable data on efficacy (reduced mortality in experimental infections following treatment), animal safety (no signs of intoxication even at three times the recommended dosage), and human food safety (muscle residues of the antibiotic). These findings contribute to the regulatory process for approving new veterinary products.

We acknowledge that analyzing residue dosage in other tissues could provide further insights into the pharmacology of the molecule and constitutes a limitation of our study. Additionally, future studies assessing the product’s efficacy under field conditions and outside controlled environments could offer valuable information on its performance in response to variables such as animal density, climate changes, and water quality.

## 5. Conclusions

In this study, florfenicol, especially when used as soon as clinical signs were detected (metaphylactic treatment), effectively reduced mortality rates in fish experimentally challenged with three main pathogens commonly found in fish farms. Furthermore, it was found that the residual concentration of the drug in muscle tissue persists for a short period, is eliminated quickly after stopping the treatment, and is below the limits established by national and international regulations. Based on these results, we conclude that the florfenicol-based medicine evaluated can be used to treat the main diseases that affect fish farming and that it is also a safe molecule for use in animals for human consumption, as evidenced by the short withdrawal period in muscle tissue. Furthermore, it is essential to emphasize that antibiotics must be used responsibly, only as a treatment for established diseases, to highlight the importance of having trained personnel to monitor livestock, and to identify and intervene early when there is a need for treatment. This way, it is possible to make rational use of these molecules in the concept of One Health.

## Figures and Tables

**Figure 1 microorganisms-13-00558-f001:**
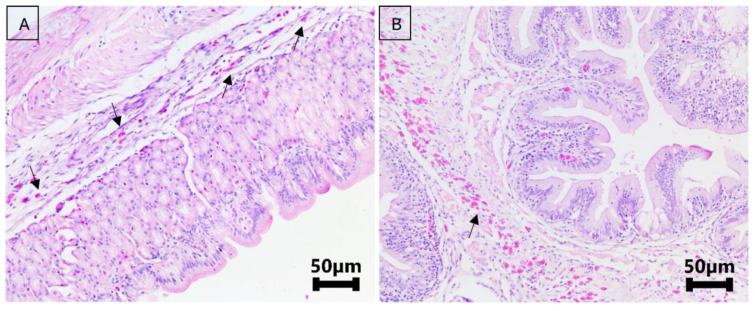
Digestive system of Nile tilapia treated and untreated with florfenicol experimentally. (**A**): Stomach of Nile tilapia from the negative control group (no treatment) showing a slight eosinophilic infiltrate in the submucosal layer (score 1) (black arrow). (**B**): Transition between the stomach and duodenum of Nile tilapia treated with florfenicol (10 mg/kg biomass) at 28 °C. A slight presence of eosinophils (score 1) is observed, particularly in this region, compared to the rest of the stomach and esophagus (black arrow). The epithelium is intact, with a well-visible mucus layer.

**Figure 2 microorganisms-13-00558-f002:**
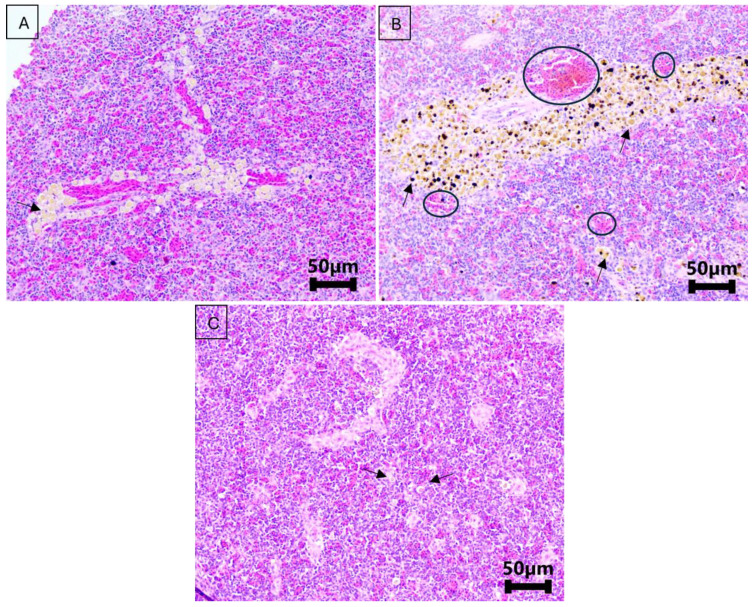
Spleen of Nile tilapia untreated and treated with florfenicol experimentally, during and after cessation of the treatment. (**A**): Spleen of Nile tilapia from the negative control group (no treatment), showing no congestion (score 0) and a slight presence of pigmented macrophages (score 1), indicated by the black arrow. (**B**): Spleen of Nile tilapia on the fifth day of treatment with florfenicol (10 mg/kg biomass) at 28 °C. The black arrow indicates a moderate accumulation of pigmented macrophages (score 2), and the circle shows areas of moderate congestion (score 2). (**C**): Spleen of Nile tilapia 10 days post treatment with florfenicol (10 mg/kg biomass) at 28 °C, showing no congestion (score 0) and a slight presence of pigmented macrophages (score 1), indicated by the black arrows.

**Figure 3 microorganisms-13-00558-f003:**
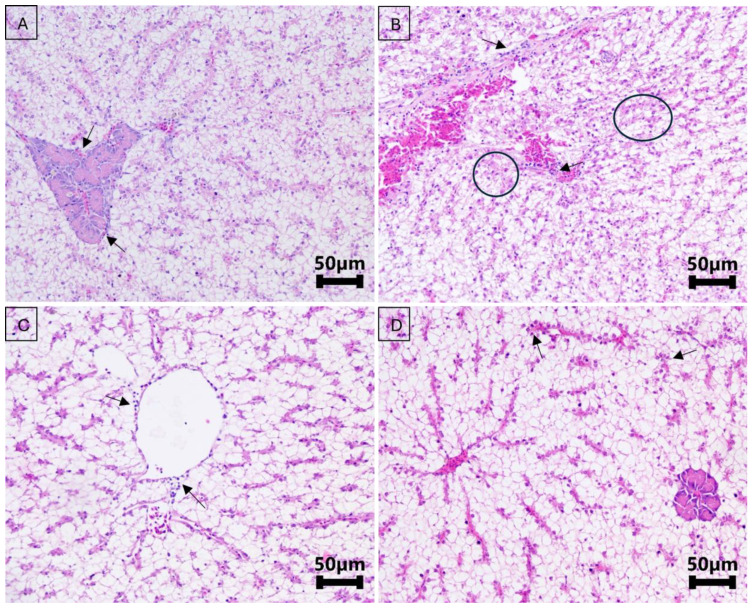
Hepatopancreas of Nile tilapia treated and untreated with florfenicol experimentally. (**A**): Negative control group (no treatment) showing mild hepatic inflammation (score 1), indicated by the black arrow, and significant glycogen accumulation in the hepatocytes (score 1). (**B**): Group treated with a florfenicol overdose on the fifth day of the treatment (30 mg/kg biomass) at 21 °C, showing a slight reduction in glycogen accumulation (score 2), mild necrosis (indicated by the black circle), and mild inflammation (black arrow) (score 1). Within the circled areas, pyknotic and karyorrhectic nuclei can be observed. (**C**): Group treated with a florfenicol overdose on the fifth day of the treatment (30 mg/kg biomass) at 28 °C, showing mild perivascular inflammation (score 1), indicated by the black arrow. (**D**): Group treated with florfenicol (10 mg/kg biomass) on the fifth day post treatment, showing only mild inflammation (score 1), indicated by the black arrow.

**Table 1 microorganisms-13-00558-t001:** Mortality rates in fish subjected to experimental infection and treated with florfenicol using three strategies. Different letters indicate statistically significant differences in the challenge × treatment comparison (*p* ≤ 0.05), with the letter “a” representing the treatment with the highest mortality.

	Type of Treatment			
Bacteria	Positive Control	Prophylactic	Metaphylactic	Therapeutic
*S. agalactiae* Ib	88/100 ^a^	21/100 ^c^	27/100 ^bc^	41/100 ^b^
*S. agalactiae* III	61/100 ^a^	2/100 ^c^	18/100 ^b^	24/100 ^b^
*F. orientalis*	57/100 ^a^	5/100 ^b^	6/100 ^b^	13/100 ^b^

**Table 2 microorganisms-13-00558-t002:** Macroscopic lesions observed in the surviving fish treated with florfenicol following the experimental challenge with *Streptococcus agalactiae* serotypes Ib and III and *Francisella orientalis*.

		Macroscopic Lesions		
Challenged Groups		Exophthalmos	Meningeal Congestion	Spleen Granuloma
*S. agalactiae* serotype Ib	PC	0/12 (0%)	2/12 (16.7%)	NI
	PT	17/79 (21.5%)	38/79 (48.1%)	NI
	MT	19/73 (26.0%)	9/73 (12.3%)	NI
	TT	19/59 (32.2%)	28/59 (47.5%)	NI
*S. agalactiae* serotype III	PC	8/39 (20.5%)	6/39 (15.4%)	NI
	PT	41/9 8(41.8%)	33/9 8(33.7%)	NI
	MT	38/82 (46.3%)	40/82 (48.8%)	NI
	TT	37/76 (48.7%)	29/76 (38.2%)	NI
*F. orientalis*	PC	NI	NI	28/43 (65.1%)
	PT	NI	NI	74/95 (77.9%)
	MT	NI	NI	73/94 (77.7%)
	TT	NI	NI	68/87 (78.2%)

PC: positive control; PT: prophylactic treatment; MT: metaphylactic treatment; TT: therapeutic treatment; NI: no injuries.

**Table 3 microorganisms-13-00558-t003:** Re-isolation of bacteria from the surviving fish treated with florfenicol in an experimental challenge to access its status as a carrier of the pathogen.

		Challenged Fish Tissue		
Challenged Groups		Brain	Kidney	Spleen
*S. agalactiae* serotype Ib	PC	3/12 (25.0%)	3/12 (25.0%)	NE
	PT	11/79 (13.9%)	20/79 (25.3%)	NE
	MT	27/73 (37.0%)	27/73 (37.0%)	NE
	TT	27/59 (45.8%)	32/59 (54.2%)	NE
*S. agalactiae* serotype III	PC	5/39 (12.8%)	0/39 (0%)	NE
	PT	2/9 8(2.0%)	2/9 8(2%)	NE
	MT	7/82 (8.5%)	2/82 (2.4%)	NE
	TT	4/76 (5.3%)	3/76 (3.9%)	NE
*F. orientalis*	PC	NE	NE	27/43 (62.8%)
	PT	NE	NE	74/95 (77.9%)
	MT	NE	NE	72/94 (76.6%)
	TT	NE	NE	68/87 (78.2%)

PC: positive control; PT: prophylactic treatment; MT: metaphylactic treatment; TT: therapeutic treatment; NE: not evaluated.

**Table 4 microorganisms-13-00558-t004:** Residual florfenicol + florfenicol-amine (μg/kg) in fish muscular tissue during and after treatment with the medicated feed.

Sample	RL (Recommended Dose, Low Temperature—21 °C)	RH (Recommended Dose, High Temperature—28 °C)
Day 1—treatment	1308.10 ± 277.80	1797.00 ± 146.90
Day 5—treatment	1148.80 ± 288.59	1896.00 ± 337.84
Day 10—treatment	1245.85 ± 272.85	1497.80 ± 322.36
Day 1—after treatment	562.60 ± 99.20	566.40 ± 134.89
Day 3—after treatment	≤500	≤500
Day 5—after treatment	≤500	≤500
Day 10—after treatment	≤500	≤500
Day 15—after treatment	≤500	≤500

## Data Availability

Data are contained within the article and Appendix A.

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
