# Peer review of "Evaluation of a Florfenicol Formulation for Treating Streptococcosis and Francisellosis in Nile Tilapia (Oreochromis niloticus): A Study of Safety, Withdrawal Period and Efficacy"

_microorganisms, 2025, doi:10.3390/microorganisms13030558_

Round 1
Reviewer 1 Report
Comments and Suggestions for Authors
Prophylactic antibiotic use in global aquaculture is widely criticized for its role in accelerating AMR, a critical One Health concern. While the manuscript presents this approach as effective in reducing mortality, it lacks necessary context regarding broader implications. Although the link between prophylactic antibiotic use and AMR is briefly acknowledged, the manuscript fails to address these broader concerns, undermining its scientific rigor and promoting practices inconsistent with global One Health principles. This oversight significantly diminishes the study’s applicability and relevance.
The study measures florfenicol residues exclusively in muscle tissue, omitting data from critical organs such as the liver, kidney, and spleen, despite histopathological evidence of lesions in these tissues. These organs are essential for understanding the drug’s pharmacokinetics and toxicological impacts. The omission of residue data from these key organs constitutes a significant gap, rendering the safety assessment incomplete. Additionally, the histological descriptions are vague and imprecise. While the manuscript references scoring criteria and descriptions in Supplementary Table S1, the supplementary data are not included, leaving critical information inaccessible. The manuscript also downplays histopathological effects as minor or reversible without providing evidence to support long-term safety or organ recovery. This lack of robust discussion on adverse effects undermines the study’s reliability and raises questions about its conclusions regarding florfenicol safety.
Moreover, the study’s exclusive reliance on laboratory conditions, without validation in field settings, limits its utility. Field trials are necessary to account for environmental variables such as water quality, pathogen exposure, and fish stress, which influence drug efficacy and safety. The absence of field validation diminishes the practical significance of the findings and restricts their applicability.
Reviewer 2 Report
Comments and Suggestions for Authors
This manuscript presents an evaluation of a new florfenicol formulation for the treatment of streptococcosis and francisellosis in Nile tilapia, focusing on its safety, withdrawal period, and efficacy. The study is well-structured and provides comprehensive data on the experimental design, methodology, results, and conclusions. However, there are several points that could be improved to enhance the quality and clarity of the manuscript.
1. The introduction section effectively provides background information on the importance of treating bacterial infections in aquaculture, particularly streptococcosis and francisellosis in Nile tilapia. However, it would be beneficial to include a brief overview of previous studies on florfenicol use in aquaculture, specifically highlighting any gaps or limitations in the existing knowledge that this study aims to address.
2. Provide more information on the selection criteria for the fish used in the study, including their health status and size distribution.
3. Describe the method used for preparing the medicated feed, including the calculation of the feeding rate and the mixing process to ensure uniformity of the florfenicol formulation in the feed.
Round 2
Reviewer 1 Report
Comments and Suggestions for Authors
The authors have made efforts to address my previous comments and clarify their arguments. While the revisions have satisfactorily resolved some concerns, further improvements in discussion section are needed to enhance the manuscript’s quality.
- The text states that only two antibiotics are approved for use in aquaculture: tetracycline and florfenicol. However, the citation [4] is unclear. Additionally, how does florfenicol compare to other approved antibiotics for tilapia infections in Brazil?
- Address the potential environmental risks of florfenicol residues in aquaculture effluents. How might its use contribute to antibiotic contamination?
- Acknowledge the study's limitations and outline potential future research directions, such as field trials and multi-drug resistance surveillance.
Author Response
Please, see the attachment.
